# Winter Is (Not) Coming: Is Climate Change Helping *Drosophila suzukii* Overwintering?

**DOI:** 10.3390/biology12070907

**Published:** 2023-06-25

**Authors:** Sara Sario, José Melo-Ferreira, Conceição Santos

**Affiliations:** 1Biology Department, Faculty of Sciences, University of Porto, 4169-007 Porto, Portugal; 2LAQV-REQUIMTE, Faculty of Sciences, University of Porto, 4050-453 Porto, Portugal; 3CIBIO-Research Centre in Biodiversity and Genetic Resources, InBIO Associate Laboratory, 4485-661 Vairão, Portugal; 4BIOPOLIS Program in Genomics, Biodiversity and Land Planning, CIBIO, 4485-661 Vairão, Portugal

**Keywords:** spotted-wing drosophila, overwinter, climate change, survival adaptation, cold acclimatation, transcriptomic changes

## Abstract

**Simple Summary:**

Drosophila suzukii, also known as the spotted-wing drosophila (SWD), is a polyphagous insect pest of soft-skinned small fruits. SWD, similar to other insects, is affected by climate change-associated factors, yet its impacts on the pest regarding its behavior, distribution, and survival remains poorly understood. Current climate change is allowing this species to colonize colder regions. This review explores how SWD adapts to survive during cold seasons, focusing on a plethora of overwintering strategies, and the transcriptomics changes in response to cold. Finally, it is discussed how climate change progression may promote the ability of this species to survive and spread, and what mitigation measures could be employed to overcome cold-adapted *D. suzukii*.

**Abstract:**

Anthropogenic challenges, particularly climate change-associated factors, are strongly impacting the behavior, distribution, and survival of insects. Yet how these changes affect pests such as *Drosophila suzukii*, a cosmopolitan pest of soft-skinned small fruits, remains poorly understood. This polyphagous pest is chill-susceptible, with cold temperatures causing multiple stresses, including desiccation and starvation, also challenging the immune system. Since the invasion of Europe and the United States of America in 2009, it has been rapidly spreading to several European and American countries (both North and South American) and North African and Asian countries. However, globalization and global warming are allowing an altitudinal and latitudinal expansion of the species, and thus the colonization of colder regions. This review explores how *D. suzukii* adapts to survive during cold seasons. We focus on overwintering strategies of behavioral adaptations such as migration or sheltering, seasonal polyphenism, reproductive adaptations, as well as metabolic and transcriptomic changes in response to cold. Finally, we discuss how the continuation of climate change may promote the ability of this species to survive and spread, and what mitigation measures could be employed to overcome cold-adapted *D. suzukii*.

## 1. Introduction

Temperature plays an important role in the overall biology of insects. As poikilothermic organisms, the insects’ body temperature depends on the environment, which affects the species’ behavior, distribution, development, and reproduction [1,2]. Due to climate change and global warming, extreme events such as droughts or heat waves are becoming more frequent. Even though an overall trend of winter warming is predicted, harsher winters with longer cold spells are still expected in some regions [3,4]. During the winter, insects face specific abiotic and biotic challenges (Figure 1a), the low temperature being the most relevant abiotic pressure. In addition to internal ice formation and consequent desiccation, cold can induce chilling injuries due to alterations in metabolic processes and homeostasis [5], and it impacts the immune system, increasing susceptibility to pathogens and the inability to behaviorally avoid pathogens and predators [6]. Additionally, with lower temperatures, fewer food sources are available, forcing insects to cope with periods of starvation [4].

To overwinter, insects have developed several strategies to overcome the inability to metabolically generate heat, either by avoiding threatening cold temperatures (*via* migration or burrowing) or developing cold tolerance [7,8]. Cold-tolerant insects are usually classified in three main types: (1) chill susceptible, (2) freeze-avoidant (or freeze-intolerant), and (3) freeze-tolerant (Figure 1b) [9]. Chill-susceptible insects die when exposed to cold temperatures without internal ice formation. The freeze-avoiding insects can survive cold as long as they can maintain their fluids in a supercooled state, and eventually die when ice formation occurs. Freeze-tolerant species can survive cold temperatures even if their body fluids freeze, but usually can only do so over a specific temperature range [7].

Typical winter low food availability and internal body ice formation lead to desiccation. Yet, low temperatures can eventually benefit freeze-tolerant insects, as they reduce the metabolic rate, allowing survival in desiccating conditions [5]. The decrease in the metabolic activity of freeze-tolerant insects is one of the physiological adaptations involved in diapause, which can either be obligate or facultative and is characterized by a suppression of development, suspended activity, and increased resistance to cold. The success of overwintering in many insect species thus depends on diapause, which can be activated during all life stages, such as eggs, pupae, larvae, nymphs, or adults [1]. These strategies to survive cold periods are triggered by temperature, together with photoperiod or humidity cues. Such cues cause flexible changes at various levels, such as molecular, transcriptomic, hormonal, metabolic, and phenotypic, which allow for overwintering and then a post-diapause recovery [10,11].

Understanding how insects cope and survive across winters in a world impacted by climate change is fundamental not only from a conservation point of view, but also because the changing climatic dynamics may influence the population trends of invasive pest species. The spotted-wing Drosophila (SWD), Drosophila suzukii (Matsumura), is an insect pest of soft-skinned fruits with a preference for small red berries, but with the capacity to thrive on a wide variety of host species, both cultivated and non-cultivated [12]. There are currently no efficient treatments to control this pest, and available agrochemicals face sustainability or efficacy limitations [13,14]. New biocontrol strategies have been proposed and include identifying new local predators [15,16] and parasitoids [17,18], among other strategies [19]. The difficulty of managing *D. suzukii* populations and the damage they cause are aggravated by the fact that this pest is highly polyphagous. *Drosophila suzukii* has diverse food and reproductive sources available, and although it shows some preferences, it can shift between hosts depending on their availability [20]. Further, wild-caught *D. suzukii* during winter months has shown the capacity of this pest to overwinter, and understanding such a process may be key to developing efficient control measures. These studies are particularly of interest, as they have already proven that overwintered individuals are the primary source of infestation during the early fruiting season [21]. 

This review goes through alternative strategies that allow *D. suzukii* to survive the cold, also addressing different behavioral adaptations such as migration strategies and sheltering. It also addresses this species’ polyphenism and reproductive adaptations, as well as metabolic and transcriptomic shifts in response to cold (Figure 2). Finally, it discusses how climate change is impacting the overwintering capacity of the species and may be promoting the survival and spread of this pest, concluding with recommendations for control/preventive measures to avoid further impacts caused by cold-adapted *D. suzukii*.

## 2. Migration and Sheltering in Non-Crop Habitats

The capacity of *D. suzukii* females to feed and lay eggs in a wide variety of cultivated and non-cultivated host species allows this pest to have access to reproduction sites and feeding sources year-round [22,23,24,25]. We have recently demonstrated that the nutrition source modulates *D. suzukii*’s energetic pathways, in a way dependent on the fruits’ nutritional geometry and sex, and that females showed higher adaptability in their energetic metabolism shift to the diet [25]. Energetically more suitable hosts may provide better conditions for feeding and development success of the offspring, and support *D. suzukii* host preferences [26,27]. However, this dependence of energetic pathways on the nutritional source may be important in winter, when food is limited and may justify the higher plasticity of females to wider energetic plasticity.

Migration between crop and non-crop habitats has already been observed, depending on the climate and host availability [28,29,30]. When temperatures start to decrease, there is an increase in *D. suzukii* individuals captured in non-crop habitat traps, such as hedgerows or forests [29]. Non-crop habitats are usually rich in wild ornamental fruit types, with different ripeness stages (from unripe to fermenting or damaged fruits), providing *D. suzukii* adults with food and oviposition sources [28,31,32]. Furthermore, as hedgerows and forest habitats are usually more dense areas, they also provide shelter from cold and desiccation, as they have higher temperatures and humidity than crop areas [33,34,35], allowing adults to enter reproductive diapause and survive the winter until the first preferred cultivated hosts start to bear fruit [21].

## 3. Seasonal Polyphenism as a Thermoregulation Strategy

The invasion success of *D. suzukii* relies not only on its wide range of host species and migration between crop and non-crop habitats, but also on its capacity to adapt to novel environments and climates through phenotypic plasticity, particularly seasonal polyphenism [36]. Seasonal polyphenism is the ability to manifest alternative phenotypes depending on annual season in response to an environmental cue. This form of phenotypic flexibility allows for coping with different environmental conditions across seasons and maintaining adaptation year-round. Seasonal polyphenism exists in a wide diversity of animals, such as changes in pattern and color in butterflies [37], or the change from brown pelage or plumage color in the summer to white in the winter in some birds and mammals (e.g., the rock ptarmigan or the snowshoe hare) [38,39]. In *D. suzukii*, seasonal polyphenism occurs as a response to an environmental temperature cue, resulting in two different phenotypes depending on the season: a summer morphotype (SM), which is the most recurrent, and a winter morphotype (WM), found in colder seasons [40].

Tran et al. (2020) established a decision tree-based method to morphometrically distinguish seasonal morphs. Both WM males and females are larger than their SM counterparts, with higher wing length, wing width, and hind tibia length [41,42]. This increase in body size in ectotherms when development temperatures are lower is common across different genera and is likely due to benefits associated with heat absorption [5]. Furthermore, larger body sizes allow for increased storage of energy sources such as sugars and fats [43]. In addition to size, *D. suzukii* seasonal polyphenism is also observed as a difference in body color: WM adults have darker cuticles, with significantly higher abdominal melanization than SM, especially on the third (in males) and fourth (in females) abdominal segments [44]. This increase in body melanization is in accordance with the theory of thermal melanism, in which darker morphotypes have an advantage over lighter individuals in conditions with lower temperatures and low levels of solar radiation, as they will eventually heat faster than the lighter morphotypes [45].

*Drosophila suzukii* life cycle is closely related to temperature and photoperiod [46]. Similar to other insects, *D. suzukii* is chill-susceptible and can rapidly suffer at temperatures below its freezing point [47,48,49]. Winkler et al. (2021) [50] estimated that the minimum temperature for *D. suzukii* females’ oviposition is 13.2 °C, and for the eggs to successfully develop into adults is 14.1 °C (although with a success rate of ≈20% compared to 85% success rate when flies were kept at the optimum development temperature). At lower temperatures, larvae and pupae also have longer developmental times, and adults can take approximately one month to emerge at 9 °C [44]. The biochemical mechanisms associated with phenotypic trait expression are often induced early in the development stages, and as the development times are longer, WM adults are usually larger than SM [51].

## 4. Transcriptomic Changes Underlying Cold Adaptation and Survival

To date, only a few studies have explored the transcriptomic differences underlying the observed seasonal polyphenism of *D. suzukii*, or how these differences can influence this pest’s management. In 2016, Shearer et al. used RNA sequencing to identify transcriptomic changes underlying WM physiology. This study showed that most of the upregulated genes in the WM were involved in metabolic pathways such as glucose metabolism, tricarboxylic acid (TCA) cycle, and glycogen metabolism. Genes involved in morphogenesis, development, and pigmentation were also upregulated in WM, explaining the phenotypic differences between the two seasonal morphs related to size and melanization. In contrast, genes involved in reproduction/oogenesis, such as those associated with the chorion, eggshell formation, or oogenesis were downregulated in the WM [44]. These differences in transcript levels were also estimated by Toxopeus et al. (2016) when analyzing the differences in the expression of genes between WM and SM by RT-qPCR, with WM showing higher levels of stress-related gene transcripts (Catalase (Cat), Superoxide dismutase (Sod), heat shock proteins (HSP)) than SM flies, as well as diapause regulation (*cpo* and *foxo*), with no expression of the Yp1 gene (vitellogenesis) detected in WM females [52]. In another study, Eriquez and Colinet (2019) acclimated 5-day-old adults to cold at 10 °C from 2 h to 9 days and showed that cold-acclimated flies had improved cold survival when compared to non-acclimated flies. Cold-acclimated and non-acclimated flies were in a chill coma after being exposed at 0 °C for 12 h, however, flies acclimated at 10 °C for 9 days had the fastest chill coma recovery time compared to the other acclimation regimes. When comparing the transcriptome of adults acclimated at 10 °C for 9 days to the non-acclimated adults, they found several upregulated and downregulated genes in the cold-acclimated flies. Most of the upregulated genes had GO-terms associated with ion transport, important to maintain ion and water homeostasis; GO-terms associated with neuronal activity and carbohydrate metabolism were also enriched. In contrast, and in accordance with the findings of Shearer et al. (2016) and Toxopeus et al. (2016), there was a downregulation of genes associated with oogenesis, suggesting that at 10 °C females will likely enter reproductive arrest [44,52,53].

The identification of these physiological changes that allow the survival of *D. suzukii* at colder temperatures is of the utmost importance not only to understand the thermal plasticity of this species, but also to develop SWD-focused management strategies. In fact, differences in the effect of insecticides between SM and WM have been suggested, especially Spinetoram, with WM appearing to be less susceptible than SM [54]. Additionally, monitoring traps are less efficient during the winter, which could be associated with differences in the response to volatiles between SM and WM [55]. Schwanitz et al. (2022) found several differentially expressed genes between SM and WM associated with the olfactory behavior of *D. suzukii*, namely those linked to food-seeking and mating behaviors, reinforcing previous work that suggested that WM flies have different food preferences than SM [56,57]. Considering that the flies surviving the winter are the most concerning at the beginning of the fruiting season [21], it is essential to understand these transcriptomic differences between SM and WM, to develop management practices targeted at WM flies and therefore reduce the impact of the winter survivors on the first fruits.

## 5. Reproductive Arrest in Winter Morphs as a Result of A metabolic Trade-Off

As the transcriptomic studies showed, the downregulation of several genes associated with reproduction in WM when compared to SM points to a reproductive arrest of WM flies. In the field, females with immature ovaries start to appear in traps during winter, also with fewer eggs than females captured in summer [46,58]. This is a result of delayed ovarian development at cold temperatures and could also be the result of a shift in the females’ metabolism when faced with adverse conditions. Additionally, more than 50% of males captured in the same period do not have sperm in their testes, but females usually already have sperm stored in their spermathecae ready to be used as females exit their reproductive arrest phase [59].

High energy reserves are necessary for females to invest in oogenesis and reproduction; however, at cold temperatures, these energy reserves are required to survive and instead of being invested in reproduction, there is a metabolic shift towards survival and extending adult lifespan [60,61]. Cold-acclimated *D. suzukii* individuals have an increased accumulation of cryoprotectant molecules such as carbohydrates, polyols, or amino acids [62], which is in line with the increase in the expression of genes associated with metabolic pathways found in transcriptomic studies, therefore revealing a metabolic trade-off of survival vs. reproduction. When *D. suzukii* females enter a reproductive diapause, they are capable of surviving longer periods and being more cold-tolerant than flies not entering reproductive diapause [58]. This reproductive diapause is then rapidly reverted, as observed by Cloutier et al., 2022, as flies that entered diapause were capable of reproducing successfully with non-diapause flies when temperatures became warmer [63].

## 6. How Is Climate Change Affecting *D. suzukii* Survival?

The impact of climate change on insect species is complex and difficult to isolate given its interdependence with many other anthropogenic stressors, such as insecticides, agricultural intensification, or deforestation, according to Wagner et al. (2021) [64]. Diverse factors are changing the population dynamics of agricultural insect pests, including the globalization of trade, which accelerates the epidemiological distribution of exotic pests; the controversial use of agrochemicals, often also affecting endemic or beneficial insects or promoting resistance in the pest population; and particularly the ongoing climate change, which tends to provide conditions to widen the geographic distribution of pests [64]. However, climate change impacts not only agricultural insect pests but also their hosts, as temperature changes eventually affect pest–host relationships and the relations of the pests with other insects, such as natural enemies [1]. The global increase in temperature during winter is one of the most important marks of climate change, even though in some regions winters are predicted to become harsher and with sudden cold spells [3,4]. In *D. suzukii*, the injury associated with cold-stress increases as temperatures decrease. However, cold acclimation improves cold tolerance to moderate or even intense cold-stress exposures [48,53,65], and their survival is especially improved when flies are exposed to fluctuating thermal regimes, which often occur during more temperate winters [66]. When exposed to 3 °C for 20 h and a small daily warm period (1 h20 at 25 °C), with some gradual cooling and warming mimicking natural environments, *D. suzukii* flies do not enter reproductive arrest and the warmer periods allow them to recover from desiccation [67]. With such a complex cold adaptation, as well as rapid cold-stress recovery, we can predict that warmer winters will help *D. suzukii* thrive in regions where it is now established, with a decreasing need to overwinter and enter reproductive diapause. Such conditions will allow to produce a higher number of generations per year [68], which may dramatically increase the demographic potential of the species and promote population growth. Additionally, warming winters in regions that are currently inhospitable for the species may promote new colonization of these regions and increase the invasive capacity of the species [69,70]. A wider area of *D. suzukii* presence worldwide and higher demographic potential will therefore lead to more fruit losses and economic damage.

## 7. Conclusions

Climate change provides a new paradigm regarding *D. suzukii*, as it promotes the dissemination of this pest in regions previously considered not ideal for its development, whilst promoting further adaptations to survive in new environmental conditions. Moreover, temperature fluctuations could have an impact on natural enemies (such as nematodes and arthropods), reducing their activity and impairing the control of *D. suzukii*.

Knowing that *D. suzukii* WM respond differently to the olfactory cues present in traps optimized for SM, together with their demonstrated higher resistance to insecticides, it is of great importance to develop new strategies focused on WM such as traps with different attractive compounds to be applied during winter (off-season) periods. This can prevent a higher incidence of this pest during producing seasons, whilst promoting a reduction in the use of harmful insecticides. Furthermore, producers should be informed and advised by phytosanitary agents to take into account these adaptations and maintain some IPM strategies year-round (e.g., mass trapping, attract-and-kill). Further studies should be conducted to understand the overwintering adaptations of *D. suzukii* and double down on the research of new specific traps/compounds and ecological management strategies to control the pest during winter periods, with emphasis on WM.

## Figures and Tables

**Figure 1 biology-12-00907-f001:**
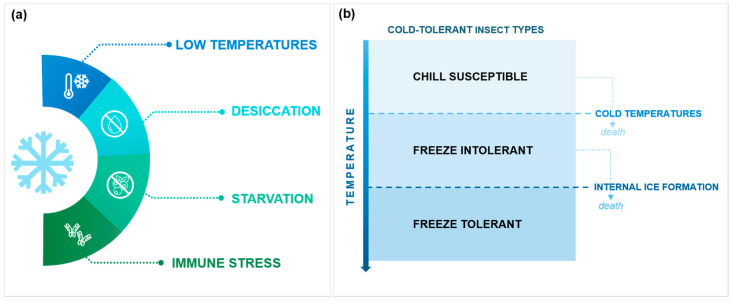
Biotic and abiotic challenges of winter (**a**) and cold-tolerant insect types (**b**).

**Figure 2 biology-12-00907-f002:**
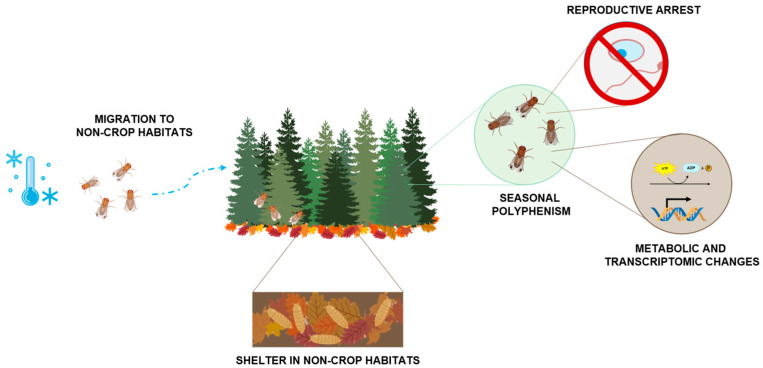
Drosophila suzukii overwintering strategies. When the temperature starts to drop, adult flies take shelter in non-crop habitats, using non-crop hosts as food and reproduction sources; immature stages such as pupae often take shelter in leaf litter. The adults that are capable of emerging after cold exposure have developed seasonal polyphenism, being larger and darker (winter morphotypes), as well as exhibiting reproductive arrest and changes at the metabolic and transcriptomic levels.

## Data Availability

No new data were created or analyzed in this study. Data sharing is not applicable to this article.

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
