# Peer review of "Winter Is (Not) Coming: Is Climate Change Helping Drosophila suzukii Overwintering?"

_biology, 2023, doi:10.3390/biology12070907_

Round 1

Reviewer 1 Report

Overall, this is a good review. Sections 4 and 5 are a bit repetitive with respect to the tradeoff between survival and reproduction. These two sections can be merged with no loss overall message. 

Reviewer 2 Report

This review, titled "Adaptation strategies of Drosophila suzukii to survive cold seasons: a review," presents recent advancements in our understanding of the effects of temperature changes on the behavior, migration, and survival of the spotted-wing fruit fly, Drosophila suzukii. The authors first outline the challenges associated with comprehending the mechanisms behind D. suzukii's cold adaptation in response to climate change. They then explore behavioral adaptations, such as migration and sheltering, that enable the species to locate suitable overwintering sites. Additionally, the authors discuss seasonal polyphenism, reproductive adaptations, and metabolic changes that contribute to the species' ability to cope with cold temperatures.

Overall, this review provides valuable insights into how D. suzukii responds and adapts to cold temperatures. Furthermore, it underscores the need for further research to mitigate the potential spread and crop damage caused by this pest in the context of ongoing climate change. Prior to publication, I have the following suggestions and comments:

1.       Figure 1b requires further modification as the link between insect survival strategies and this figure is not apparent.

2.       Figure 2 also needs some corrections. Firstly, the inclusion of adult flies, in addition to larval and pupal stages, in shelter non-crop habitats is necessary. Secondly, please ensure that the color and size differences in seasonal polyphenism are clearly discernible.

3.       It would be helpful to include a table summarizing all the genes and metabolic changes discussed in sections 4 and 5, specifically focusing on winter in comparison to other seasons.

4.        provide a reference for lines 104-105.

5.       In line 146, the authors mention that the life cycle length is related to temperature and photoperiod, but I was unable to find supporting information regarding photoperiod.

6.       Please provide a reference for lines 104-105.

7.       Please provide a reference for lines 258-359.

I didn't notice any obvious Grammar mistakes, but few sentences are wordy.

Reviewer 3 Report

This is the comprehensive report written by the authors, covering the effects of climate change, particularly winter stress on the target pest. 

This manuscript looks fine and may be acceptable for publication.

Reviewer 4 Report

Congratulations aunthors is a good review.

Reviewer 5 Report

Interesting review. Please, incorporate more relevant bibliography, not just about D. suzukii but also research on other Drosophila spp.

All changes suggestions are pointed out in the attached file

Improve grammar and writing. See some suggestions regarding this matter in the attached document.

Round 2

Reviewer 5 Report

Manuscript in good shape, ready for publishing.